

# Lipopolysaccharide induces a downregulation of adiponectin receptors *in-vitro* and *in-vivo*

Alison Hall[1], Martin Leuwer[2], Paul Trayhurn[3,4,5] and Ingeborg D. Welters[6,7]

[1] Department of Critical Care, Royal Liverpool University Hospital, Liverpool, Obesity Biology Research Unit, University of Liverpool, Liverpool, United Kingdom
[2] Department of Molecular & Clinical Pharmacology, University of Liverpool, Liverpool, United Kingdom
[3] Obesity Biology Research Unit, University of Liverpool, Liverpool, United Kingdom
[4] Clore Laboratory, University of Buckingham, Buckingham, United Kingdom
[5] College of Science, King Saud University, Riyadh, Saudi Arabia
[6] Department of Ageing and Chronic Disease, University of Liverpool, Liverpool, United Kingdom
[7] Department of Critical Care, Royal Liverpool University Hospital, Liverpool, United Kingdom

Corresponding author
Ingeborg D. Welters,
ingewelt@yahoo.com

## ABSTRACT

**Background.** Adipose tissue contributes to the inflammatory response through production of cytokines, recruitment of macrophages and modulation of the adiponectin system. Previous studies have identified a down-regulation of adiponectin in pathologies characterised by acute (sepsis and endotoxaemia) and chronic inflammation (obesity and type-II diabetes mellitus). In this study, we investigated the hypothesis that LPS would reduce adiponectin receptor expression in a murine model of endotoxaemia and in adipoocyte and myocyte cell cultures.

**Methods.** 25 mg/kg LPS was injected intra-peritoneally into C57BL/6J mice, equivalent volumes of normal saline were used in control animals. Mice were killed at 4 or 24 h post injection and tissues harvested. Murine adipocytes (3T3-L1) and myocytes (C2C12) were grown in standard culture, treated with LPS (0.1 μg/ml–10 μg/ml) and harvested at 4 and 24 h. RNA was extracted and qPCR was conducted according to standard protocols and relative expression was calculated.

**Results.** After LPS treatment there was a significant reduction after 4 h in gene expression of adipo R1 in muscle and peri-renal fat and of adipo R2 in liver, peri-renal fat and abdominal wall subcutaneous fat. After 24 h, significant reductions were limited to muscle. Cell culture extracts showed varied changes with reduction in adiponectin and adipo R2 gene expression only in adipocytes.

**Conclusions.** LPS reduced adiponectin receptor gene expression in several tissues including adipocytes. This reflects a down-regulation of this anti-inflammatory and insulin-sensitising pathway in response to LPS. The trend towards base line after 24 h in tissue depots may reflect counter-regulatory mechanisms. Adiponectin receptor regulation differs in the tissues investigated.

## INTRODUCTION

White Adipose Tissue (WAT) is now known to be a dynamic secretory organ in its own right, secreting a number of compounds called adipokines (*Robinson, Prins & Venkatesh, 2011*). These biologically active proteins act as inflammatory mediators and play a major role in metabolic derangements in chronic inflammatory disorders such as type 2 Diabetes mellitus (DM) and the metabolic syndrome (*Kadowaki et al., 2006*; *Kern et al., 2003*). Newer research has demonstrated inhibition of anti-inflammatory adipokines in acute inflammatory processes such as severe sepsis (*Welters et al., 2014*). Similar responses are also observed after lipopolysaccharide (LPS) challenge, which therefore provides a useful model for the study of altered metabolism in inflammation (*Agwunobi et al., 2000*).

First described in the early 2000s, Adiponectin is a 30 kDa, 244-amino acid polypeptide, which is mainly expressed in adipose tissue and has anti-inflammatory, anti-diabetic and anti-atherogenic effects. It has a similar structure to complement factor C1q and accounts for 0.01% of total plasma protein (*Whitehead et al., 2006*). Adiponectin increases glucose utilisation and reduces insulin resistance by stimulating fatty acid oxidation which in turn leads to reduced triglyceride concentration in skeletal muscle and liver (*Fruebis et al., 2001*; *Yamauchi et al., 2002*). The anti-inflammatory effects of adiponectin include suppressed proliferation of myeloid cell lines, reduction of the phagocytic ability of macrophages and down-regulation of macrophage recruitment to sites of inflammation (*Tsuchihashi et al., 2006*; *Yokota et al., 2000*). Adiponectin also reduces the production of inflammatory cytokines from macrophages and adipose tissue (*Park et al., 2008*; *Tsuchihashi et al., 2006*; *Yokota et al., 2000*). Observations of adiponectin in chronic disease such as type II DM, obesity and cardiovascular disease identify consistent down-regulation of gene and protein expression (*Hu, Liang & Spiegelman, 1996*; *Maeda et al., 2002*; *Robinson, Prins & Venkatesh, 2011*). In acute inflammation, preliminary human studies have confirmed a similar down-regulation of adiponectin (*Welters et al., 2014*) and small animal studies have demonstrated a negative correlation with pro-inflammatory cytokine concentrations such as Tumour Necrosis Factor-$\alpha$ (*Bruun et al., 2003*).

Two adiponectin receptors have been identified, adipoR1 and R2 (*Yamauchi et al., 2003*). Both are expressed in numerous tissues including skeletal muscle, liver, adipose tissue and pancreatic islet and acinar cells (*Civitarese et al., 2004*; *Kharroubi et al., 2003*; *Tsuchida et al., 2004*). Previous studies have identified a down-regulation of adiponectin and its receptors in pathologies characterised by chronic inflammation such as obesity and type-II DM (*Kadowaki & Yamauchi, 2005*; *Tsuchida et al., 2004*). In a previous study, we demonstrated the down-regulation of adiponectin gene and protein expression within different fat depots 24 h following LPS administration in mice (*Leuwer et al., 2009*). These results are in line with reports on decreased circulating adiponectin levels in the plasma of septic patients (*Hillenbrand et al., 2010*; *Uji et al., 2009*). However, to date, little is known about the modulation of the adiponectin system in peripheral organs involved in glucose and lipid metabolism, such as liver and skeletal muscle. It is tempting to speculate that global acute inflammation elicits similar changes to the adiponectin system not only within the adipose tissue itself, but also in peripheral tissues involved in lipid and glucose metabolism.

In this study, we investigated the hypothesis that LPS reduces adiponectin receptor expression in a murine model of endotoxaemia and also in mouse fat and muscle isolated cell lines.

## MATERIALS AND METHODS

### Animal experiments

All experiments were carried out on 8 to 10-week-old male C57BL/6J mice (Charles River, Oxford, UK). All experimental procedures were approved by the UK Home Office and were conducted in accordance with the appropriate Project License (PPL 40/2692). Mice were housed in separate cages post procedures and maintained in the same temperature-controlled conditions ($22 \pm 2$ °C, 12 h light/12 h dark cycle) with free access to a standard laboratory rodent diet and water. LPS (25 mg/kg, Escherichia coli O 111:B4, Sigma-Aldrich) was injected intra-peritoneally (ip) under general anaesthesia (2% isoflurane in $N_2O/O_2$). All animals received 1 ml of normal saline subcutaneously (SC) at time of LPS injection to compensate for fluid losses. Control animals received an equivalent volume of normal saline i.p. Both control and LPS treated mice were killed at 4 ($n = 6$) and 24 h ($n = 9$), respectively, after injection by cervical dislocation. Based on recommendations by the UK Home Office, sample sizes were reduced to the minimum number expected to yield significant results. Similar studies required <10 animals per group to demonstrate significant changes (*Leuwer et al., 2009*). Peri-renal fat (PRF), epididymal fat (EF), abdominal wall subcutaneous fat (SCF), skeletal muscle (soleus muscle) and liver were removed and immediately frozen in liquid nitrogen until analysis.

### Cell culture

Isolated cell lines were used to investigate LPS effects on fat and muscle cells (3T3-L1 murine adipocytes and C2C12 murine myocytes). Cells were initiated in culture media (Dulbecco's modified Eagle medium (DMEM) (Sigma-Aldrich, Gillingham, Dorset, UK) with 10% foetal calf serum (FCS) (3T3-L1 murine adipocytes) and 10% FCS with 1% penicillin/streptomycin and L-glutamine (C2C12 murine myocytes). Cells were incubated at 37 °C, in a humidified atmosphere of 95% air and 5% $CO_2$ until confluence was reached. 3T3-L1 adipocytes were differentiated by the addition of 10 mg/ml insulin, 1 mM dexamethasone, and 100 mM IBMX in DMEM. C2C12 myocytes were differentiated with 2% horse serum. Cells were treated with different concentrations (0.1, 1, 5, 10 μg/ml) of LPS (Escherichia coli O 111:B4, Sigma-Aldrich, Gillingham, Dorset, UK) and harvested at 4 and 24 h. Control experiments were performed using equivalent volumes of normal saline. Each experiment was repeated at least six times. Untreated control cells were harvested at the same time points.

### RNA extraction and real time PCR

RNA extraction and reverse transcription were performed as previously described (*Leuwer et al., 2009*). Briefly, total RNA was extracted from adipose tissues with Trizol reagent (Invitrogen, UK), and 1 μg of DNase I-treated RNA was reverse transcribed using a Reverse-iT$^{TM}$ 1$^{ST}$ Strand Synthesis Kit (Abgene, Epsom, UK) in the presence of anchored

oligo dT in a total volume of 20 μl. Real-time PCR was conducted using TAQ Man (12.5 μl reaction volume with 12.5 ng of cDNA with optimal concentrations of primers and probes and qPCR$^{TM}$ Core kit (Eurogentec, UK) (Beta actin: Forward: ACGGCCAGGTCAT-CACTATTG, Reverse: CAAGAAGGAAGGCTGGAAAAG, Adiponectin R1: Forward: AGATGGAGGAGTTCGTGTA TAAGG, Reverse: GGCCATGTAGCAGGTAGTCG, Adiponectin R2: Forward: CTTTCGGGCCTGTTTTAAGAGC, Reverse: ATATTTGGGC-GAAACATATAAAAGATCC, Adiponectin: Forward: GGCTCTGTGCTCCTCCATCT, Reverse: AGAGTCGTTGACGTTATCTGCATAG). All qPCR reactions were analysed with the housekeeping gene $\beta$-actin.

## Statistical analysis

Relative gene expression levels were determined using the 2-ddCt method (*Livak & Schmittgen, 2001*). Data are presented as mean values ± Standard Error of Mean. Differences between groups were analyzed by Student's unpaired *t*-test or non-parametric tests when data was non-normally distributed. In the animal model, treatment 4 h and 24 h after LPS injection was compared with a respective control group. Results were considered to be statistically significant when $p < 0.05$. Where multiple comparisons were performed in the *in vitro* study, statistical significance was corrected using Bonferroni's method for each time point. Fold change was calculated as 1/2-ddCt.

# RESULTS

## Adiponectin receptors are down-regulated in murine endotoxaemia

Expression of adiponectin receptors was detected in all murine tissues examined. 4 h after treatment with LPS, there were significant reductions in adipoR1 gene expression in skeletal muscle (9.8 fold reduction, $p = 0.017$) and peri-renal fat (PRF) (1.6 fold reduction, $p = 0.008$) (Table 1A and Data S2). AdipoR2 gene expression decreased in liver (2.7 fold reduction, $p = 0.008$), PRF (4.3 fold reduction, $p = 0.004$) and sub-cutaneous fat (SCF) (2.9 fold reduction, $p = 0.04$). This represents a rapid response to treatment with LPS. In mice treated with LPS for 24 h, there were significant reductions in the expression of both receptors in skeletal muscle (adipoR1: 1.9 fold reduction, $p = 0.01$ and adipoR2 2.2 fold reduction, $p = 0.05$) (Table 1B).

## Cell-type specific downregulation of adiponectin receptors

To identify whether the effects of LPS *in vivo* reflect a direct effect of LPS treatment on specific cell types, we examined relevant cell lines, using LPS as a direct stimulus in cell cultures. We found that adiponectin receptor gene expression differed in adipocytes and myocytes: in adipocytes, receptor down-regulation was restricted to adipoR2 at higher doses of LPS (1 and 10 μg/ml) (2.5 fold reduction, $p = 0.02$ and 3.9 fold reduction, $p = 0.01$ respectively). This change was observed only in cells treated for 4 h (Fig. 1) while those treated for 24 h (Fig. 2) exhibited no response.

In myocytes, only a minimal change in receptor gene expression following treatment with LPS was detectable. There was a small, but statistically significant down-regulation in

**Table 1 Adiponectin receptor gene expression in murine tissue depots.** Relative change adiponectin receptor gene expression in mouse tissue depots 4 h (A) and 24 h (B) after treatment with LPS 25 mg/kg. Gene expression was determined by real-time PCR. Relative gene expression was calculated using the 2-ddCt method and $p < 0.05$ was considered significant. The reference group for calculations was the control group (ip saline injection) and housekeeping gene was $\beta$-actin.

| 4 h | AdipoR1 | | AdipoR2 | |
| --- | --- | --- | --- | --- |
| (A) | | | | |
| | 2-ddCt | *p* value | 2-ddCt | *p* value |
| Control | 1 | | 1 | |
| Liver | 0.544 | 0.05 | 0.371 | 0.008[*] |
| Muscle | 0.102[*] | 0.017[*] | 0.162 | 0.39 |
| EF | 0.671 | 0.48 | 0.543 | 0.24 |
| PRF | 0.627[*] | 0.0087[*] | 0.231 | 0.0043[*] |
| SCF | 0.821 | 0.81 | 0.348 | 0.041[*] |

| 24 h | AdipoR1 | | AdipoR2 | |
| --- | --- | --- | --- | --- |
| (B) | | | | |
| | 2-ddCt | *p* value | 2-ddCt | *p* value |
| Control | 1 | | 1 | |
| Liver | 0.614 | 0.09 | 0.650 | 0.148 |
| Muscle | 0.509[*] | 0.01[*] | 0.448 | 0.05[*] |
| EF | 1.01 | 0.47 | 0.852 | 0.55 |
| PRF | 0.801 | 0.27 | 0.657 | 0.198 |
| SCF | 0.824 | 0.47 | 1.058 | 0.98 |

**Notes.**
[*] $p < 0.05$.
SEM, standard error of mean; EF, Epididymal fat; PRF, Peri-renal fat; SCF, Subcutaneous fat.

adipoR1 after treatment with 5 μg/ml LPS for 4 h ($p = 0.02$) (Fig. 3) and small increases in adipoR2 after 24 h ($p < 0.01$) (Fig. 4).

## Adiponectin gene expression is downregulated early after LPS stimulation

Adiponectin receptor down-regulation was accompanied by a dose-dependent reduction in adiponectin gene expression in 3T3-L1 adipocytes. This downregulation was only observed 4 h (Fig. 5) after stimulation with LPS, while 24 h after treatment no significant changes were found (Fig. 6). This is in line with previous reports showing a normalisation of adiponectin gene expression 24 h after treatment of mice with LPS (*Leuwer et al., 2009*).

## DISCUSSION

LPS induces an acute inflammatory response in most organs, including WAT, liver and skeletal muscle. These changes extend to several metabolic pathways, including the adiponectin system. Thus far, the down-regulation of adiponectin and its receptors has been well described in chronic conditions associated with low-grade inflammation such as obesity and type II DM (*Kadowaki et al., 2006*; *Kern et al., 2003*). Chronic low-grade inflammation leads to elevated concentrations of pro-inflammatory cytokines such as

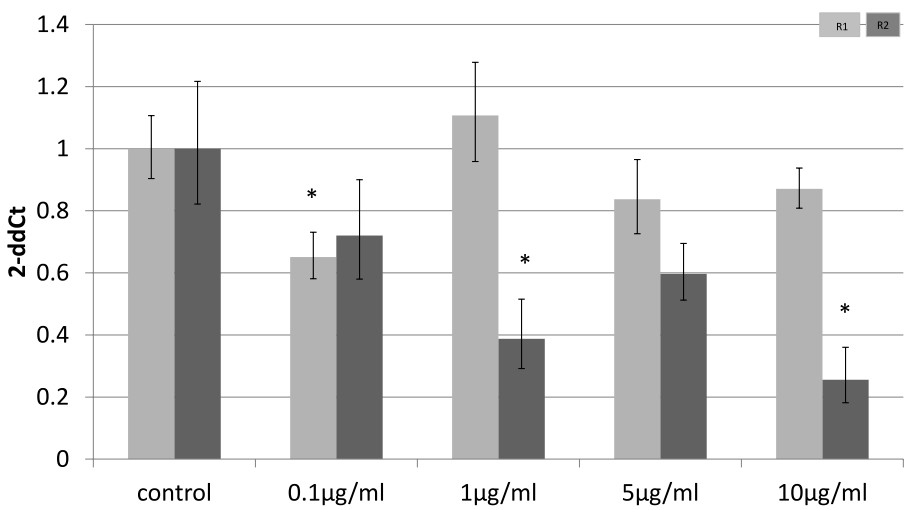

**Figure 1 Adiponectin receptor gene expression in 3T3-L1 adipocytes 4 h following LPS challenge.** Relative change adiponectin receptor gene expression in mouse adipocyte cell line (3T3-L1) incubated for 4 h ($n = 6$ run in duplicate) with varying concentrations of LPS. Gene expression was determined by real-time PCR. Relative gene expression was calculated using the 2-ddCt method and $p < 0.05$ was considered significant. Error bars correspond to Standard Error of Mean (*$p < 0.05$, **$p < 0.01$). Housekeeping gene was $\beta$-actin. (R1, AdipoR1; R2, AdipoR2).

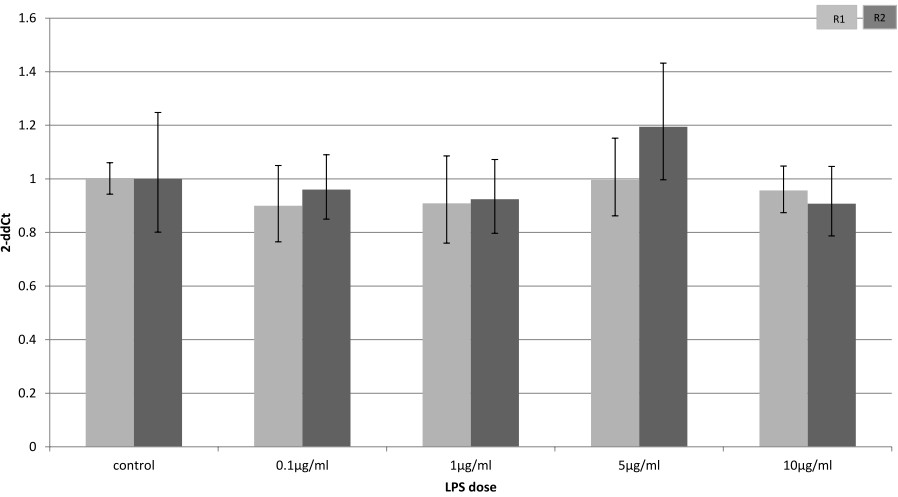

**Figure 2 Adiponectin receptor gene expression in 3T3-L1 adipocytes 24 following LPS challenge.** Relative change adiponectin receptor gene expression in mouse adipocyte cell line (3T3-L1) incubated for 24 h ($n = 6$ run in duplicate) with varying concentrations of LPS. Gene expression was determined by real-time PCR. Relative gene expression was calculated using the 2-ddCt method and $p < 0.05$ was considered significant. Error bars correspond to Standard Error of Mean (*$p < 0.05$, **$p < 0.01$). Housekeeping gene was $\beta$-actin. (R1, AdipoR1; R2, AdipoR2).

TNF-$\alpha$ and Interleukin-6 (IL-6) which may play a role and both mediators suppress adiponectin production (*Fantuzzi, 2008*). Our results demonstrate that adiponectin receptor gene expression is altered in response to LPS challenge. To the best of our knowledge, change in adiponectin receptor gene expression in response to an acute LPS challenge has not been investigated before in a mouse model of severe sepsis.

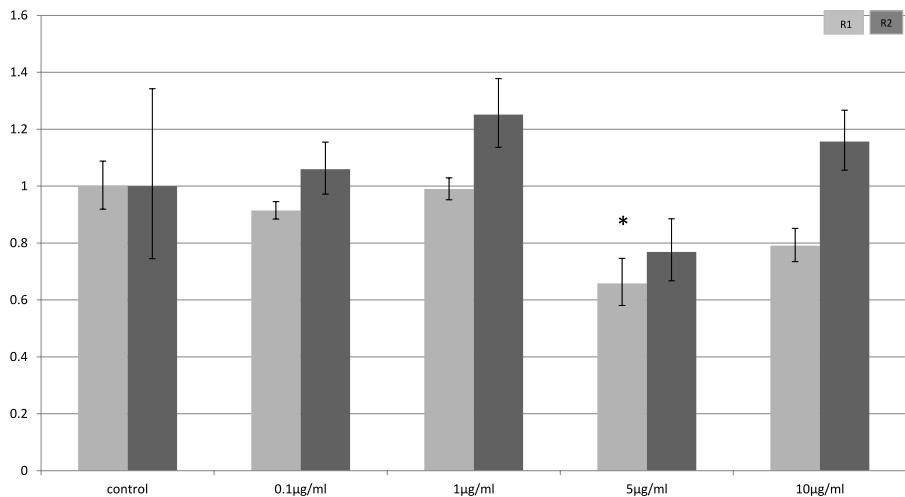

**Figure 3 Adiponectin receptor expression in C2C12 myocytes 4 h following LPS challenge.** Relative change adiponectin receptor gene expression in mouse isolated myocytes (C2C12) incubated for 4 h ($n = 6$ run in duplicate) with varying concentrations of LPS. Gene expression was determined by real-time PCR. Relative gene expression was calculated using the 2-ddCt method and $p < 0.05$ was considered significant. Error bars correspond to Standard Error of Mean (*$p < 0.05$, **$p < 0.01$). Housekeeping gene was $\beta$-actin. (R1, AdipoR1; R2, AdipoR2).

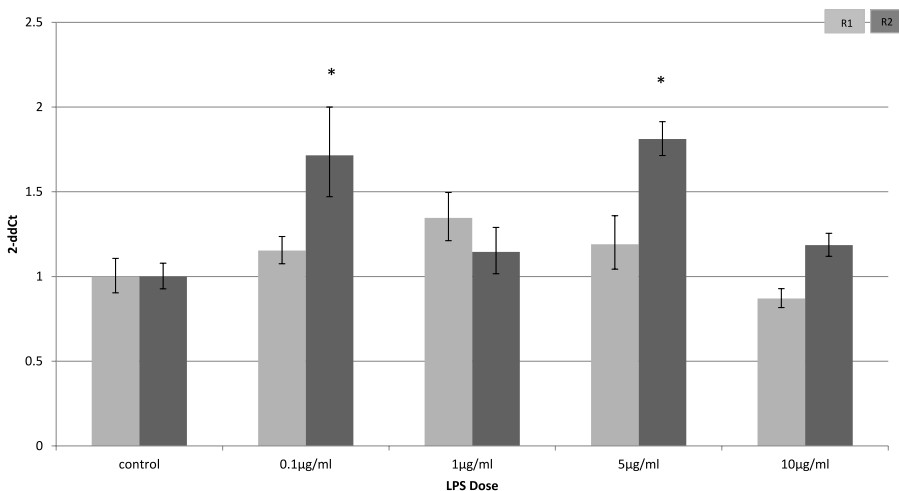

**Figure 4 Adiponectin receptor expression in C2C12 myocytes 24 h following LPS challenge.** Relative change adiponectin receptor gene expression in mouse isolated myocytes (C2C12) incubated for 24 h ($n = 6$ run in duplicate) with varying concentrations of LPS. Gene expression was determined by real-time PCR. Relative gene expression was calculated using the 2-ddCt method and $p < 0.05$ was considered significant. Error bars correspond to Standard Error of Mean (*$p < 0.05$, **$p < 0.01$). Housekeeping gene was $\beta$-actin. (R1, AdipoR1; R2, AdipoR2).

In acute inflammatory processes, adipose tissue responds to systemic endotoxaemia in a similar fashion by producing early rises in inflammatory cytokine expression, in particular IL-6 and TNF-$\alpha$. In endotoxaemic mice, these changes have been demonstrated to be accompanied by reduced adiponectin gene and protein expression in multiple depots of adipose tissue (*Leuwer et al., 2009*). We further support these findings by demonstrating

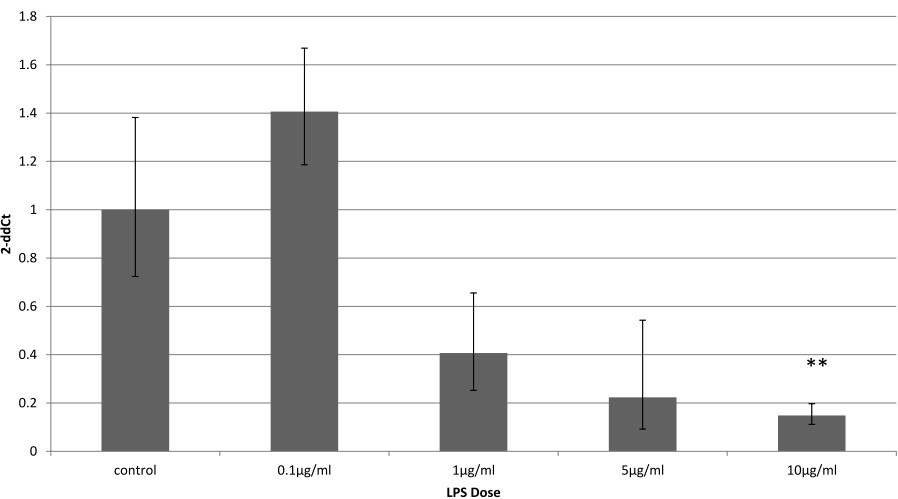

**Figure 5 Adiponectin gene expression in 3T3-L1 adipocytes 4 h following LPS challenge.** Relative change adiponectin receptor gene expression in mouse adipocyte cell line (3T3-L1) incubated for 4 h ($n = 6$ run in duplicate) varying concentrations of LPS. Gene expression was determined by real-time PCR. Relative gene expression was calculated using the 2-ddCt method and $p < 0.05$ was considered significant. Error bars correspond to Standard Error of Mean ($*p < 0.05$, $**p < 0.01$). Housekeeping gene was $\beta$-actin. (R1, AdipoR1; R2, AdipoR2).

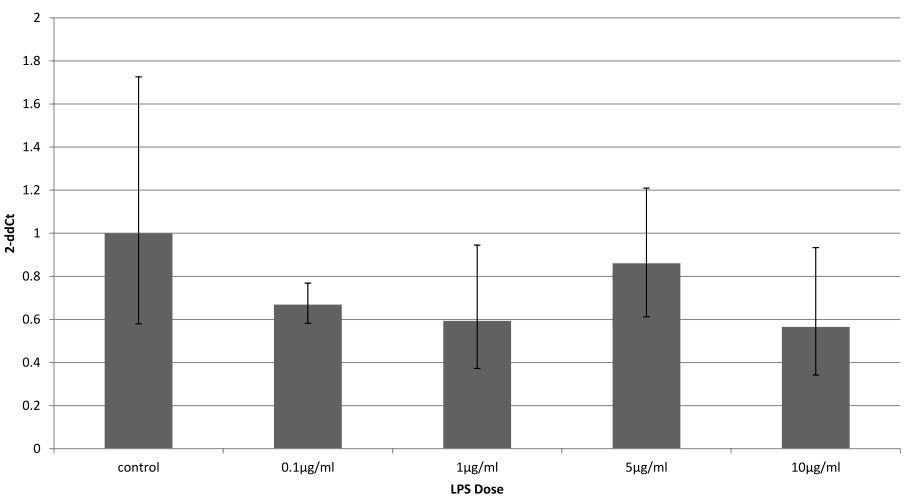

**Figure 6 Adiponectin gene expression in 3T3-L1 adipocytes 24 h following LPS challenge.** Relative change adiponectin receptor gene expression in mouse adipocyte cell line (3T3-L1) incubated for 24 h ($n = 6$ run in duplicate) varying concentrations of LPS. Gene expression was determined by real-time PCR. Relative gene expression was calculated using the 2-ddCt method and $p < 0.05$ was considered significant. Error bars correspond to Standard Error of Mean ($*p < 0.05$, $**p < 0.01$). Housekeeping gene was $\beta$-actin. (R1, AdipoR1; R2, AdipoR2).

adiponectin gene expressions down-regulation in cultured 3T3-L1 adipocytes. However in a recent report, the use of lower LPS doses in female rats has produced conflicting results: concentrations of adiponectin in visceral and subcutaneous WAT remained unchanged or even increased (*Iwasa et al., 2014*). Gender differences as well as the use of low LPS (5 mg/kg compared to 25 mg/kg LPS in our study) may contribute to this discrepancy.

Similarly, in human volunteers, intravenous administration of low dose endotoxin has failed to reduce circulating adiponectin despite acute rises in inflammatory cytokines (*Anderson et al., 2007*; *Keller et al., 2003*). Since mild endotoxaemia may neither reflect severe LPS responses nor human sepsis, appropriately used a high-dose LPS model to induce a peracute activation of the immune system.

The mouse model used in this study represents severe sepsis and is known to produce sharp but transient increases in pro-inflammatory cytokines accompanied by severe reductions in cardiac output and blood pressure (*Dyson & Singer, 2009*; *Remick & Ward, 2005*). Although human sepsis differs from experimental endotoxaemia, a study investigating the adiponectin system in human sepsis identified significantly lower mean circulating adiponectin in septic patients, which supports the concept that the adiponectin system is downregulated in acute infection and inflammation (*Venkatesh et al., 2009*).

Our animal model demonstrated a significant but depot-dependent down-regulation of adiponectin receptors in adipose tissue, skeletal muscle and liver. The time points were chosen to demonstrate early changes and also potential normalisation occurring at 24 h. The response from adipose tissue varied depending on the depot investigated. In perirenal (visceral) fat, both adiponectin receptor subtypes were down-regulated, while in subcutaneous fat only adipoR2 down-regulation was observed. There were no changes in epididymal fat depots. This may reflect redistribution of tissue perfusion following the inflammatory insult or could alternatively result from the metabolic differences between visceral and subcutaneous fat (*Bergman et al., 2007*; *Nannipieri et al., 2007*).

Receptor down-regulation in fat depots and adipocyte cell line cultures was transient. In adipocytes, we demonstrated a rapid onset dose-dependent reduction in adipoR2 gene expression within 4 h of LPS treatment which was accompanied by reduced adiponectin mRNA levels. Interestingly, there was no change in adipoR1 expression. This is in contrast to previous results which demonstrated down-regulation of both receptors by Staphylococcus Aureus-derived peptidoglycans after only 3 h of treatment in 3T3-L1 adipocytes (*Ajuwon, Banz & Winters, 2009*), despite staphylococcal proteins being capable of inducing cytokine release by 3T3-L1 adipocytes (*Vu et al., 2013*).

*In vivo*, factors other than LPS contribute to the down-regulation of the adiponectin system in early acute inflammation. It has previously been demonstrated that insulin has an inhibitory effect on adipoR1 expression in 3T3-L1 adipocytes (*Inukai et al., 2005*). Therefore, high insulin levels associated with endotoxaemia and infection may have influenced the down-regulation of adipoR1 gene expression observed in our study (*Fasshauer et al., 2002*). Adiponectin itself also regulates adiponectin receptor expression (*Mistry et al., 2006*) and could have affected expression in our *in vivo* model. However, the decrease in adipoR2 gene expression in 3T3-L1 adipocytes precedes changes in adiponectin gene expression and therefore supports the concept that LPS induces down-regulation of this receptor subtype by mechanisms independent of adiponectin. The role of other metabolic changes including insulin resistance, acidosis and hypoxia remain to be investigated in this context.

In skeletal muscle, there was a prolonged effect of LPS on adiponectin receptor expression. This effect again was limited to *in-vivo* experiments and not seen in the myocyte cell line (C2C12). It is known that skeletal muscle produces myokines during exercise and under inflammatory conditions when glycogen stores are low (*Pedersen, 2009*; *Pedersen & Febbraio, 2008*). Myokines, including IL-6 and other pro-inflammatory cytokines, exert paracrine effects locally on the skeletal muscle as well as endocrine effects when they are released into the systemic circulation (*Pedersen, 2009*; *Pedersen & Febbraio, 2008*). The lack of endocrine effects in a cell culture model may account for the discrepancy between *in-vivo* and *in-vitro* models.

Evidence suggests that adiponectin receptors may represent two distinct entities (*Bluher et al., 2006*). AdipoR1 deficient mice have been shown to have impaired glucose tolerance, insulin resistance and increased endogenous production of glucose (*Yamauchi et al., 2007*), while adipoR2 knock-out mice are lean, resistant to diet-induced obesity, weight gain and hepatic steatosis, and display reduced plasma cholesterol and lower fasting insulin. However, their glucose tolerance is impaired as demonstrated by increased plasma insulin concentrations (*Bjursell et al., 2007*; *Yamauchi et al., 2007*). This indicates that receptor regulation may be tissue-specific and subtype specific.

While adipoR1 is ubiquitously expressed, including abundant expression in skeletal muscle, adipoR2 is most abundantly expressed in the liver (*Kadowaki et al., 2006*). Regulation of adipoRs in the liver is less well described than in skeletal muscle, but there is some evidence that PPARα agonists increase adipoR expression (*Tsuchida et al., 2005*). Furthermore, incubation of hepatocytes or myocytes with insulin reduces the expression of adipoR1 and adipoR2 (*Tsuchida et al., 2004*), indicating that insulin may play a direct role in adiponectin receptor expression regulation. Thus, insulin resistance associated with systemic LPS challenge may be involved in the adiponectin receptor down-regulation observed in the liver extracts in our *in vivo* model. Two other studies have confirmed the down-regulation of adipoR2 in the hepatic tissue of mildly endotoxaemic male and female rats (*Iwasa et al., 2014*; *Sakai et al., 2013*) but without change in adipoR1 expression.

Our study is limited in that we did not determine circulating adiponectin, glucose or insulin levels in the *in-vivo* model. Hence, further experiments to investigate the relation between insulin, glucose and triglyceride levels and adiponectin receptor expression are required. In addition, the translation of changes in gene expression into changes in protein measurement requires to be elucidated. Although a trend towards normalisation in adiponectin receptor expression was demonstrated after 24 h, our experiments only provide results for the early stages of endotoxaemia. Further investigations including longer term consequences of adiponectin receptor down-regulation in systemic inflammation are therefore warranted. Our results only allow conclusions for male animals, gender differences in adiponectin receptor expression have been described in a previous report (*Iwasa et al., 2014*). The comparison of LPS doses from animal experiments to cell lines is difficult as response to external LPS challenge varies between cell lines. Thus, we accept that there are limitations in comparing the doses of LPS used in both experimental setups.

## CONCLUSIONS

Taken together, the down-regulation of adiponectin receptors in muscle, liver and fat depots in endotoxaemia may contribute to insulin resistance and hyperglycaemia frequently found in clinical conditions associated with acute inflammation. The trend towards normalisation of adiponectin receptor expression after 24 h *in vivo* may reflect activation of counter-regulatory mechanisms within the body to limit the pro-inflammatory response and the metabolic derangements. This counterregulation is unlikely to represent clinical improvement as all the mice continued to display overwhelming symptoms of acute inflammation. Cell culture systems may lack the capacity for effective control of LPS effects, which could explain the longer duration of adiponectin receptor down-regulation under *in vitro* conditions.

**List of abbreviations**

| | |
|---|---|
| **LPS** | Lipopolysaccharide |
| **DM** | Diabetes Mellitus |
| **WAT** | White adipose tissue |
| **KDa** | Kilodaltons |
| **adipoR1** | Adiponectin receptor 1 |
| **adipoR2** | Adiponectin receptor 2 |
| **Ip** | Intra-peritoneal |
| **$N_2O/O_2$** | Nitrous oxide/oxygen mix |
| **PRF** | Peri-renal fat |
| **EF** | Epididymal fat |
| **SCF** | Sub-cutaneous fat |
| **DMEM** | Dubecco's modified eagles medium |
| **FCS** | Fetal calf serum |
| **qPCR** | Quantitative Polymerase chain reaction |
| **IL-6** | Interleukin-6 |
| **TNF-$\alpha$** | Tumour necrosis factor-alpha |
| **PPAR-$\alpha$** | Peroxisome proliferator-activated receptor-alpha |

## ACKNOWLEDGEMENT

We are very grateful to Mr. Leif Hunter who helped with the experiments.

### Funding

The authors received no funding for this work.

### Competing Interests

The authors declare there are no competing interests.

## Author Contributions

- Alison Hall conceived and designed the experiments, performed the experiments, analyzed the data, contributed reagents/materials/analysis tools, wrote the paper, prepared figures and/or tables, reviewed drafts of the paper.
- Martin Leuwer and Paul Trayhurn conceived and designed the experiments, contributed reagents/materials/analysis tools, reviewed drafts of the paper.
- Ingeborg D. Welters conceived and designed the experiments, analyzed the data, contributed reagents/materials/analysis tools, reviewed drafts of the paper.

## Animal Ethics

The following information was supplied relating to ethical approvals (i.e., approving body and any reference numbers):

Experimental procedures were approved by the UK Home Office and were conducted in accordance with the appropriate Project License (PPL 40/2692).

## Supplemental Information

Supplemental information for this article can be found online at http://dx.doi.org/10.7717/peerj.1428#supplemental-information.

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
