# Peer review of "Lipopolysaccharide induces a downregulation of adiponectin receptors in-vitro and in-vivo"

_PeerJ, doi:10.7717/peerj.1428_

## Round 0.1 · original submission · Major Revisions

· Academic Editor

Major Revisions

I wonder if you could address some of the concerns of reviewer 2. It seems to me that the suggestions given would greatly increase the visibility and impact of the study.

·

Basic reporting

All figure legends should be corrected to represent only the incubation time shown in the actual figure ( separate 4 and 24 h).

Table 1 is really hard to decipher. A more standard presentation would probably help. Also, the legend needs to report the exact reference group that was used for calculations of DD ct.

Experimental design

See below regarding power analysis.

Validity of the findings

The investigators may want to include an explanation of how they determined group size and number of replications for their experiments. Support of those parameter are by power analysis would highly increase statistical robustness of the findings.

Reviewer 2 ·

Basic reporting

The paper adds interesting information to the current knowledge of LPS regulation of adiponectin and its receptors. However, most of the results reported are confirmatory of previous published observations and this should be of interest for PeerJ readers.
However, There are certain flaws that should amended.
I would like to suggest some points that I hope authors will take into consideration.
The introduction is poor, as well discussion. The role of adiponectin in response to inflammatory stimuli is poorly discussed , This reviewer suggests to introduce and discuss on the both the roles of adiponectin in inflammatory response. To this regard, a number of good reviews are suggested to be cited:
Robinson et al. Critical Care 2011, 15:221;
Drug Discov Today. 2014 Mar;19(3):241-58.
J Allergy Clin Immunol. 2008 Feb;121(2):326-30.
Basic Clin Pharmacol Toxicol. 2014 Jan;114(1):97-102.

Experimental design

Experiments carried on in cultured cell lines should be mandatory reproduced in mouse primary cultures of adipocytes and miocytes.
Most of the results obtained by authors should be discussed considering those reported in Endocr J 61:891-900.
Other minor points
The first paragraph of discussion (line 169) should be deleted /changed. The statement is not correct.
3T3-L1 is a cell line (not isolated mouse cells) . Please, correct along the ms
Primary adipocytes culture are isolated cells.

Validity of the findings

One big limitation is the lack of determination of circulating adiponectin in the animal model and the lack of results in murine primary cells. . These aspects should be mandatory added to give the present findings much more strength.

Comments for the author

no comments.

---

## Round 0.2 · accepted · Accept

· Academic Editor

Accept

All the points raised by the reviewers have been satisfactorily addressed.